# The Effect of In Vitro Digestion on Polyphenolic Compounds and Antioxidant Properties of Sorghum (*Sorghum bicolor* (L.) Moench) and Sorghum-Enriched Pasta

**DOI:** 10.3390/molecules28041706

**Published:** 2023-02-10

**Authors:** Agnieszka Ziółkiewicz, Kamila Kasprzak-Drozd, Agnieszka Wójtowicz, Tomasz Oniszczuk, Marek Gancarz, Iwona Kowalska, Jarosław Mołdoch, Adrianna Kondracka, Anna Oniszczuk

**Affiliations:** 1Department of Inorganic Chemistry, Medical University of Lublin, Chodźki 4a, 20-093 Lublin, Poland; 2Department of Thermal Technology and Food Process Engineering, University of Life Sciences in Lublin, Głęboka 31, 20-612 Lublin, Poland; 3Faculty of Production and Power Engineering, University of Agriculture in Krakow, Balicka 116b, 30-149 Krakow, Poland; 4Institute of Agrophysics Polish Academy of Sciences, Doświadczalna 4, 20-290 Lublin, Poland; 5Department of Biochemistry and Crop Quality, Institute of Soil Science and Plant Cultivation, State Research Institute, 24-100 Puławy, Poland; 6Department of Obstetrics and Pathology of Pregnancy, Medical University of Lublin, Staszica 16, 20-081 Lublin, Poland

**Keywords:** in vitro digestion, sorghum, polyphenols, liquid chromatography, antioxidant activity

## Abstract

The phenol content of sorghum is a unique feature among all cereal grains; hence this fact merits the special attention of scientists. It should be remembered that before polyphenols can be used in the body, they are modified within the digestive tract. In order to obtain more accurate data on the level and activity of tested ingredients after ingestion and digestion in the in vivo digestive tract, in vitro simulated digestion may be used. Thus, the aim of this study was to determine the content of polyphenols, flavonoids, and individual phenolic acids, as well as the antiradical properties, of sorghum and sorghum-enriched pasta before and after in vitro simulated gastrointestinal digestion. We observed that the total content of polyphenols decreased after gastric digestion of sorghum, and slightly increased after duodenal digestion. Moreover, the flavonoid content decreased after the first stage of digestion, while antioxidant properties increased after the first stage of digestion and slightly decreased after the second stage. The digestion of polyphenolics in sorghum is completely different to that in pasta—both in varieties with, and without, the addition of sorghum. For pasta, the content of total polyphenols and flavonoids, and free radical scavenging properties, decrease after each stage of digestion.

## 1. Introduction

Sorghum (*Sorghum bicolor* (L.) Moench) is one of the basic cereals grown in the world today, and in ranking, it is the fifth most cultivated cereal grain [1]. It is consumed predominantly in semi-arid and subtropical regions as the main component of, among others, porridges, flatbreads, and beer, and constitutes the basic food product in these regions. Although it is a gluten-free cereal grain [2], in areas where sorghum does not form the basis of the diet, it is used as an addition to cakes, instant dishes, breakfast cereals, meatless burgers, and pretzels [1]. Sorghum can be a valuable source of protein, dietary fiber, polysaccharides, vitamins (especially B vitamins), and minerals, which vary in their quantitative content depending on various factors; for example, variety, method of processing, cooking, and cooling [3].

The phenol content of sorghum is a unique feature among cereal grains; hence this fact has drawn the attention of food scientists [4]. Polyphenolic compounds are characterized by various biological activities, e.g., inhibition of the growth of cancer cells [5]. Consuming whole grain sorghum can help reduce the incidence of many health problems, such as heart disease, obesity, and diabetes. There is a need, however, to continue scientific research on the possibility of using sorghum for health purposes [3]. These studies are part of the current trend for interest in food, which, apart from its nutritional function, can have additional pro-health benefits. Such studies have demonstrated that whole grains of sorghum have numerous benefits for human health, especially in relation to the antioxidant activity of phenolic compounds present in the outer layers of the grain [6,7]. The high antioxidant potential of compounds found in sorghum is also associated with a reduction in oxidative stress [6], and antimicrobial [8], anti-inflammatory [9] and anti-cancer properties [10]. It is believed that the beneficial effect of sorghum grains on human health is mainly related to the presence of the polyphenolic compounds contained in them. Many types of polyphenols have been identified in sorghum; for example, flavonoids, derivatives of hydroxybenzoic acid, and hydroxycinnamic acid [10,11].

The gastrointestinal digestion of foods significantly influences the bioaccessibility of biologically active compounds, such as polyphenols. Since plant foods are diverse in composition or eaten in conjunction with other foods, food bolus constituents can modulate the bioaccessibility and stability of phytochemicals. It should be remembered that before polyphenols can be used in the body, they are modified within the digestive tract in order to be assimilated. However, most research is based only on the determination of the concentration of phytonutrients in food products. These studies do not provide complete information on the actual bioavailability of these components. In order to obtain more accurate data on the level and activity of tested ingredients after ingestion and digestion in the in vivo digestive tract, in vitro simulated digestion may be used [12,13]. Thus, the aim of this study was to determine the content of polyphenolic compounds, flavonoids, and individual phenolic acids, as well the antiradical properties, of sorghum and pasta enriched with sorghum before and after in vitro simulated gastrointestinal digestion.

There are factors that can lead to concentrations in tissues well below those that exhibit antioxidant capacity in vitro. These include the complex biotransformation of plant metabolites after ingestion and their negligible bioavailability. The antioxidant potential of dietary components found in the digestive tract may play a crucial role in the body’s defense mechanism at the systemic level [14].

## 2. Results and Discussion

### 2.1. Nutrient Composition of Sorghum Grain

Cereals are staple foods providing a source of proteins, carbohydrates, fibers, vitamins, and minerals for a substantial group of the world’s population. Sorghum is rich in active phytochemicals that may have pro-health benefits. This is especially seen in populations whose diets are mainly based on plant-based food [4]. The current study supports the growing evidence for the unique health benefits of sorghum whole grain consumption.

The chemical composition of sorghum samples is shown in Table 1. The table also provides the recommended daily dosage (RDI) according to the European legislation [15].

The results obtained by the authors are consistent with those presented by Pontieri et al. [4] in terms of the content of both total proteins and fats. The above researchers analyzed the grain composition of three varieties of sorghum of different pericarp colors (white, black, and red) grown in the Mediterranean region. They observed minor differences in both protein and carbohydrate among varieties. A higher fiber content was found in both the red and black varieties compared with white pericarp. The results obtained by these authors show that sorghum whole grain flour made from grain with varying pericarp colors contains unique nutritional properties. According to the research, black sorghum showed slightly higher total protein levels and less total fat. This may be of fine-scale benefit when black sorghum flour is used in human food products. Our research showed significantly lower carbohydrate content compared with Pontieri et al., (48.49 g/100 g in our studies vs. 70.07–73.17 g/100 g in Pontieri et al.) and significantly higher levels of fiber (28.27 g/100 g in our study vs. 5.37–7.78 g/100 g in Pontieri et al.). Nyoni et al., reported that sorghum grain contains 55–75% starch and 1–6% crude fiber on a dry weight basis. It also contains some anti-nutritional factors (tannins and phytins). These bind to fiber, proteins, and other nutrients present in grain, making them unavailable for intestinal absorption, thus inhibiting their digestibility [16]. The sorghum we tested contained significantly more fiber, to the detriment of starch. It is also possible that in our case, the higher fiber content was due to some extent to the presence of phytins or tannins associated with it. The higher fiber content found in our sorghum sample suggests that this variety may have health benefits in addition to those granted by just the phenolic compounds present in these grains. But on the other hand, dietary fibers are the carriers for polyphenols and thus influence their bioaccessibility, as fiber-entrapped phenolics are both difficult to extract and barely soluble in the gastrointestinal fluids. In general, soluble and insoluble polysaccharides can bind phenolic compounds and limit their diffusion and substrate–enzyme contacts during gastrointestinal digestion [13]. For comparison, the fiber and carbohydrate content of other grains are as follows: brown rice—1 g/100 g fiber and 76 g/100 g carbohydrate, wheat—2 g/100 g fiber and 71 g/100 g carbohydrate, maize—2.8 g/100 g fiber and 73 g/100 g carbohydrate, pearl millet—2.3 g/100 g fiber and 67 g/100 g carbohydrate, and finger millet 3.6 g/100 g fiber and 72.6 g/100 g carbohydrate [16].

### 2.2. Polyphenol, Flavonoid and Free Phenolic Acid Content in Sorghum Grain

Phenolic compounds from sorghum grain are obtained through many extraction techniques [16,17], and the extraction yield, content, and profile of phenolics in sorghum vary depending on the extractant used. Devi et al. [18], for example, reported that a methanol extract of sorghum bran polyphenols showed a greater anthocyanin content (1.95 mg/g) than did an acetone extract (1 mg/g). Nowadays, emerging methods have been applied for phenolic compound extraction from sorghum, the aim of which is to enhance the derived polyphenol content. So far, ultrasound-assisted extraction [19], microwave-assisted extraction [20], pulsed-electric field, accelerated solvent extraction [21], and subcritical water extraction [22] have been explored. Accordingly, Luo et al. [19] reported that the polyphenolic content of red sorghum bran was higher using an ultrasound-assisted extraction method (UAE) at elevated temperatures, compared with conventional techniques and with an aqueous methanol method previously tested by the authors [14]. As UAE was demonstrated to be a useful method for isolating phenols (e.g., flavonoids and phenolic acids) from food, the authors of this paper used this technique.

In the first phase of the experiment, we examined the total polyphenolic content (TPC) and total flavonoid content (TFC). The results showed that sorghum is a rich source of phenolics (as gallic acid equivalent per 1 g of dry weight of the raw material: 3.620 mg GAE/g d.w.) and flavonoids (0.536 mg GAE/g d.w.). These results are consistent with those obtained by other authors who have reported 0.24–34.78 mg polyphenols (mg GAE/g d.w.) and 0.06–0.38 mg flavonoids (as rutin equivalent per 1 g of d.w. of the raw material; mg RE/g d.w.) in sorghum with white pericarp [10], 4.13–11.50 mg polyphenols (mg GAE/g d.w.) and 0.00–0.20 mg flavonoids (mg RE/g d.w.) in sorghum with black pericarp [10,23], and 3.58 mg polyphenols (mg GAE/g d.w.) in sorghum with brown pericarp [10]. Other researchers have also analyzed the TFC content of raw white sorghum varieties. Herein, the PC 5 hybrid had 0.1533 mg RE/g [24], 0.448 mg RE/g and 0.267 mg RE/g [24] depending on the solvent used for extraction. The findings of our study also show that the extracts held relatively high radical scavenger activity against DPPH—63.240%.

In most cases, the antioxidant activity of plant extracts was found to be strongly correlated with their TPC [21]. Dicko et al., for example, presented the results of the TPC and biological activities of 50 sorghum varieties before and after germination. While some cultivars exhibited an increase in the TPC and antioxidant activity after germination, others had a slight decrease. In general, the TPC was determined to be at 880 mg GAE/100 g before and 920 mg GAE/100 g after germination [25].

The next step of the analysis was to assess free phenolic acid content. We used reversed-phase ultra-high pressure liquid chromatography equipped with a photodiode array detector and coupled to a triple-quadrupole mass spectrometer. The following phenolic acids were identified in the samples: protocatechuic (3.663 µg/g d.w.), *p*-OH-benzoic (6.947 µg/g d.w.), vanillic (1.612 µg/g d.w.), syryngic (0.107 µg/g d.w.) (benzoic acid derivatives), as well as caffeic (0.568 µg/g d.w.), and *p*-coumaric (1.461 µg/g d.w.) (cinnamic acid derivatives). The results of research conducted by other authors have also revealed the presence of cinnamic acid derivatives-caffeic and *p*-coumaric acids [26,27], while Ghinea et al. [24] noted chlorogenic, gallic, and tannic acids in sorghum extracts.

### 2.3. Effect of In Vitro Digestion on the Content of Polyphenolic Compounds, Flavonoids, and Free Phenolic Acids and Antiradical Properties of Sorghum

Before polyphenols can be used in the body, they are modified within the digestive tract in order to be assimilated [28]. 

Hydrochloric acid contained in gastric juice supports the denaturation of proteins and activates pepsin. The length of this phase is 2–4 h. Free phenolic acids are absorbed in the stomach; in addition, these compounds can be conjugated with glucuronic acid. The effect of the next stage of intestinal digestion on the bioavailability of polyphenols depends on various factors. For example, increasing the concentration of phenolic compounds may affect the matrix of intestinal enzymes. By contrast, degradation or isomerization of polyphenols may occur as a result of catalysis in the presence of oxygen. Moreover, absorption of aglycones and their glucosylated forms by the small intestine can occur by passive diffusion or active transport. In the large intestine, polyphenol compounds are degraded by the colonic bacterial flora. Depending on the structure, many different polyphenols may be formed within, but ultimately the metabolites of all these compounds lead to the formation of benzoic acid [29].

In our study, we used a two-stage, in vitro digestion model, including gastric and duodenal phases. Before carrying out in vitro digestion, we verified the total content of polyphenolics and flavonoids, the free phenolic acid content, and the antioxidant properties of the samples.

We observed that the total polyphenol content decreased after gastric digestion of sorghum and slightly increased after duodenal digestion (compared with the gastric digestion phase). Moreover, the flavonoid content decreased significantly after the first stage of digestion, while the antioxidant properties increased after the first stage of digestion and slightly decreased after the second stage (Table 2). 

Phenolic compounds are mainly found in plants in glycosylated, polymerized, and esterified forms. Thus, during digestion, they can be hydrolyzed in the acidic stomach environment, in the alkaline environment of the intestine, and by the effect of digestive enzymes. These conditions result in changes in the structure of the phenolics, e.g., glycosylation, hydroxylation, and dimerization, as well as in partial degradation of their primary structure. Bioaccessibility of polyphenols is therefore dependent on the type and on the amount present in the plant matrix [13]. In our previous work, we determined the content of polyphenolic compound and individual phenolic acids, and also the antiradical properties, of corn snacks enriched with wild garlic, before and after in vitro simulated gastrointestinal digestion. For all the investigated snacks, the concentration of polyphenols after in vitro digestion was significantly reduced after gastric and duodenal digestion, as compared with the samples before digestion. The free phenolic acid content also decreased drastically after the first stage of in vitro digestion (gastric), which was deepened during the second stage (duodenal). In addition, the authors found that the in vitro digestion procedure reduced the snack’s antioxidant activity [14].

Siracusa et al. [30] and Baeza et al. [31] have reported that the total concentration of polyphenols was reduced during the gastric digestion stage, for aqueous infusions from *Capparis spinosa* L., *Crithmum maritimum* L. [30], chamomile tea, yerba mate, coffee-like substitutes, and coffee blend [31]. We saw similar results in our study. However, unlike our research, a decrease in the content of active compounds depended upon activity within the intestinal part of in vitro digestion [30,31]. Dacrema et al. [32] also reported a drop in the polyphenolic content after in vitro digestion of fireweed extract. Accordingly, they observed a loss of individual polyphenolic compounds after the gastric phase in the range of 1.92–84.17%, and a 11.83–98.07% decrease after the duodenal phase.

Our research has shown that the antioxidant properties of sorghum increased after the first stage of digestion, as compared with the state before digestion, and then slightly decreased after the second stage. The decrease in antioxidant activity after the intestinal stage is consistent with the results presented by other authors. A similar study of the effect of in vitro digestion on antioxidant activity of model phenolic compounds demonstrated a decrease in activity for rosmarinic acid (24–36%) and caffeic acid (12–19%), and no change in the antioxidant capacity of rutin after the duodenal stage [33]. A decrease in antioxidant activity during the intestinal part of in vitro digestion was observed in our study, as well as in the work of other scientists on this topic. This may be due to the structural reorganization of some bioactive compounds when the reaction changes to a slightly alkaline state.

It should be noted that pH conditions can have an influence on both in vitro and in vivo studies of the antioxidant potential of polyphenolic compounds. This aspect is key when considering the influence of the pH of selected parts of the digestive tract on the structures and activity of plant compounds. Nonetheless, study results are limited. It is common knowledge that the antioxidant potential of plant material is correlated with the number and position of OH groups in the main compounds, as well as their hydrogen-donating abilities. Thus, in order to explain the pH influence on antioxidant activity of food rich in polyphenols, each component must be taken into account [34].

Lettuce is a plant rich in polyphenols such as flavonoids, caffeic acid derivatives, and chlorogenic acid [35]. The results of a study involving lettuce extract revealed that its antioxidant ability increased with increasing pH, while pH 7 caused a slight decrease in activity [36]. Other results were obtained for extracts of sweet potato leaf [37], a plant rich in chlorogenic acids and caffeic acids. In this study, the alkaline environment had a negative impact on the antioxidant potential, while the neutral and weakly acidic environment caused an increase in the discussed activity. Rodríguez-Roque et al. [38] subjected a soy drink rich in phenolic compounds to in vitro digestion and noticed a higher antioxidant capacity in gastric digestion as compared with intestinal digestion. Similarly, for honey, which contains flavonoids (apigenin, quercetin, luteolin, and hesperitin) and phenolic acids (benzoic, cinnamic, vanillic, coumaric, caffeic, chlorogenic, and ellagic), the authors observed significantly decreasing activity with increasing pH [39]. 

The reduction in antioxidant capacity under intestinal conditions can be attributed to the structural reorganization of some compounds due to their sensitivity to alkaline pH. Moreover, these compounds can bind to other ingredients of the matrix, resulting in the formation of complexes that may also contribute to the reduction in their antioxidant activity [13].

Our research has shown the antioxidant properties of sorghum increased after the first stage of digestion compared with the state before digestion and then slightly decreased after the second stage. The decrease in antioxidant activity after the intestinal stage is consistent with the results presented by other authors. A study of the effect of in vitro digestion on antioxidant activity of model phenolic compounds demonstrated a decrease in activity for rosmarinic acid (24–36%) and caffeic acid (12–19%) and no change in the antioxidant capacity of rutin after the duodenal stage [33]. The reduction in antioxidant activity observed in our study and in the work of other authors during the intestinal part of in vitro digestion can be explained by the structural reorganization of some bioactive compounds during a change to a slightly alkaline pH (Table 3).

Different findings were obtained by Majdoub et al. [40]. They noted a decrease in certain compounds after in vitro gastric and duodenal digestion. The level of caffeoylquinic acid was especially reduced after 20 min of gastric digestion. Similarly, coumaroylquinic acids only slightly persisted during simulated digestion as a result of degradation occurring in the gastric and duodenal compartments. Moreover, quinic acid merely existed in the samples obtained during duodenal digestion (this compound was probably derived from the degradation of the more complex caffeoylquinic and coumaroylquinic acids). No remarkable differences were observed for rutin between the initial amount and the amounts recovered during the gastric step. In black carrot, a decrease in phenolic acid content (chlorogenic, cryptochlorogenic, and neochlorogenic) was also observed during the gastric stage of in vitro digestion, which was escalated at further stages of digestion, whereas ferulic and caffeic acid demonstrated higher levels as compared with undigested samples [41].

It is not an effortless task to compare bioavailability studies because of the presence of different variables that can affect digestion in the gastrointestinal tract. The differences in the results may be due to the influence of the plant and food matrix and the heterogeneity of the analyzed plant raw materials, as well as the method of performing the in vitro digestion procedure. Pure compounds also show high variability: for example, in rutin, the % of loss after intestinal digestion was found to be from only 3% to total loss, while in the case of quercetin, the results ranged from 5.8% [42] to total loss [30], and for chlorogenic acid, from 44% to 95.7% [30]. It is thus well recognized that the digestion methodology is an important factor for assessing the bioaccessibility of polyphenols. 

### 2.4. Effect of In Vitro Digestion on the Content of Polyphenolic Compounds, Flavonoids, and Free Phenolic Acids and Antiradical Properties of Pasta Enriched with Sorghum

Sorghum consumption is known to demonstrate pro-health effects, including lowered levels of lipids and glucose in the blood. In addition, the active compounds contained in sorghum may be helpful in controlling weight by participating in mechanisms that increase satiety [43].

In patients suffering from diabetes and prediabetes, glucose disorders can induce many serious health problems. To counteract these conditions, it may be useful to follow diets in which food items have undergone appropriate food modifications that slow down the release of glucose into the blood. One example is the reduction of starch digestibility. This is achievable by creating higher levels of resistant starch in food. According to the scientific work of Lemlioglu-Austin et al. [43], an increase in the Resistant Starch (RS) content in corn porridge and a decrease in the digestibility of starch and the Estimated Glycemic Index (EGI) can come about by the addition of sorghum extracts to food items. Their research has also indicated that reduced EGI value and increased RS content in these corn groats were at a similar or higher level compared with those found in legumes, whole grain pasta, or cereals [44]. Beta et al. [45], in turn, contribute that whole sorghum porridges were characterized by significantly (*p* < 0.05) lower digestibility of starch than whole grain corn porridge.

Taking into account the results presented by these researchers, we created a durum wheat pasta enriched with sorghum and we tested this product for the content of polyphenolic compounds, flavonoids, and free phenolic acids, as well as its antiradical properties, before and after in vitro digestion. The research results showed that even a small addition of sorghum induced an increase in the polyphenol content (including flavonoids and free phenolic acids) in pasta, as well as an increase in antioxidant activity (Table 4 and Table 5). 

It is easy to notice that the digestion of polyphenolic compounds in sorghum is completely different to that in pasta—both in varieties with and without the addition of sorghum. For pasta, total polyphenols, flavonoid content, and (especially) free radical scavenging properties decrease after each stage of digestion (Table 4). In both types of pasta, the presence of three phenolic acids was determined before digestion: protocatechuic, *p*-OH-benzoic, and vanillic (Table 5). Two of these acids—protocatechuic and vanillic—were not observed after digestion, while the benzoic acid content increased successively after both stages of digestion. 

Performing the principal components analysis (PCA) allowed us to obtain five new variables, and the first three principal components explain 99.90% of the variability of the system. Figure 1a shows the projection of the variables on planes PC1 (78.58%) and PC2 (11.83%), which describe the dependencies at 90.41%.

A strong positive correlation was found between vanillic acid, protocatechuic acid, flavonoid content, and radical scavenging activity. The correlation between these parameters and *p*-OH-benzoic acid was strong and negative. The correlation between vanillic acid, protocatechuic acid, flavonoid content, radical scavenging activity, and polyphenol content is positive but weak. In turn, there is no correlation between polyphenol content and *p*-OH-benzoic acid. All compounds found in the two-circle area strongly affect the determination ability of the pasta before and after two-stage digestion and with the sorghum addition. Figure 1b shows cases of pasta before and after two-stage digestion and with the sorghum addition. Positive PC1 values describe cases before digestion, and negative PC1 values describe cases after two-stage digestion. In turn, the second principal component (PC2) in Figure 1b describes the case of pasta with the sorghum addition. Positive PC2 values describe pasta without sorghum addition, and negative PC2 values describe cases of pasta with sorghum addition.

Due to the fact that PC3 is close to the PC2 value, the third principal component (PC3) was also used in the interpretation of the analysis results (PCA). Figure 2a shows the projection of the variables on planes PC1 (78.58%) and PC3 (9.49%), which describe the dependencies at 88.07%.

A strong positive correlation was found between vanillic acid, protocatechuic acid, and flavonoid content, and radical scavenging activity. The correlation between vanillic acid, protocatechuic acid, and flavonoid content, radical scavenging activity, and polyphenol content is positive but weak. The correlation between these parameters and *p*-OH-benzoic acid was weak and negative. In turn, the correlation between polyphenol content and *p*-OH-benzoic acid is strong and negative. All compounds found in the two-circle area strongly affect the determination ability of the pasta before and after two-stage digestion and with the sorghum addition. Figure 2b shows cases of pasta before and after two-stage digestion and with the sorghum addition. Positive PC1 values describe cases before digestion, and negative PC1 values describe cases after two-stage digestion. In turn, the third principal component (PC3) in Figure 2b describes the case after two-stage digestion. Positive PC3 values describe gastric digestion and negative PC3 values describe cases of duodenal digestion.

The differences in the digestion of polyphenolic compounds contained in sorghum and those present in pasta results from the various structures of matrices. Research has demonstrated that during pasta digestion, phenolics can bind to starch [45]. Furthermore, tannins form polymers (among themselves or with other food ingredients, in particular, carbohydrates and proteins) which are more difficult to extract [46]. Tannin interactions with food ingredients are mostly non-covalent and may include hydrogen bonding and hydrophobic interactions. The tannins contained in sorghum show a strong affinity for prolamine proteins, which are high in proline [47]. 

Condensed tannins are suspected to be damaged or structurally modified during the digestion process. According to Dlamini [48], tannins that are subjected to low pH and elevated, depolymerizing temperatures are oligomers and monomers, with the basic phenolic structure remaining stable. However, upon exposure to higher pH, the condensed tannins are polymerized and cross-linked, and insoluble polymers with a higher molecular weight are formed.

Differences in the digestion of polyphenolic compounds during the digestion of sorghum and pasta are clearly visible in the correlations presented for individual groups of compounds (Table 6). However, despite the differences, for all samples, one regularity can be seen: their antioxidant properties are in high positive correlation with the level of free phenolic acids. According to previously conducted experiments, aglycones are characterized by higher antioxidant activity than their glycosidic forms and are combined by means of bonds of different types [49]. Hence, perhaps the antioxidant properties of sorghum, both after the first and second stage of digestion, are higher than before the process.

## 3. Materials and Methods

### 3.1. Chemicals

Acetonitrile and formic acid obtained for chromatographic analysis, ethanol for extraction, and Folin-Ciocalteu reagent were provided by J.T. Baker (Phillipsburg, NJ, USA). The standards of phenolics and 2,2-diphenyl-1-picrylhydrazyl (DPPH), sodium bicarbonate, hydrochloric acid, pancreatin, sodium glycodeoxycholate, sodium taurocholate, sodium taurodeoxycholate, were purchased from Sigma Aldrich (St. Louis, MO, USA). Water was purified on a MilliQ system (Millipore S.A., Molsheim, France).

### 3.2. Pasta Production

Semolina and red colored sorghum (*Sorghum bicolor* (L.) Moench) were purchased from the local Algerian market. Sorghum flour was produced from sorghum grain samples that were milled (Chopin Moulin CD1; Chopin S.A., Villeneuve la Garenne, France) and then sieved using a planetary sieve (Buhler AG, Uzwil, Switzerland, screen size 120 µm^2^).

The pasta without the addition of sorghum contained only semolina, salt, and water, while the pasta enriched with sorghum contained semolina (98%), salt, water, and sorghum (2%). Samples of pasta are presented in Figure 3. The main stages of manufacture of pasta are presented in Figure 4.

### 3.3. Extraction Procedure

The extractions of semolina and pasta samples were performed in an ultrasonic bath (Bandelin Electronic GmbH & Co. KG, Berlin, Germany) using 80% methanol for 40 min at a temperature of 60 °C in two identical cycles (ultrasound frequency of 33 kHz using 320 W of power). The extracts were filtered and combined, evaporated to dryness and dissolved in methanol [14].

### 3.4. Determination of Phenolic Acids

Determination of phenolic acids was carried out according to the modified published method of Burda and Oleszek [50]. The free phenolic acid content was assessed by reversed-phase ultra-high pressure liquid chromatography on a Waters ACQUITY UPLC^®^ Systems chromatograph (Waters Corporation, Milford, MA, USA), equipped with a photodiode array detector and coupled to a triple-quadrupole mass spectrometer (Waters ACQUITY^®^ TQD, Micromass, Manchester, UK). Samples were separated on a Waters ACQUITY UPLC^®^ HSS C18 column (1.0 × 100 mm; 1.8 μm) at 30 °C. The mobile phase contained 0.1% formic acid in water (*v*/*v*) and 0.1% formic acid in acetonitrile, (*v*/*v*). The analytes were eluted using a combination of isocratic and gradient steps.

The detection of phenolic acids was developed in the negative ionization mode, using a selected reaction monitoring method. The source temperature was 110 °C and the desolvation temperature was 350 °C. Nitrogen was used as a desolvation gas (the flow of 1000 L/h) and as a cone gas (100 L/h), while argon was used as a collision gas (0.1 mL/min). Concentrations of phenolic acids in sorghum extracts were calculated on the basis of calibration curves.

### 3.5. Determination of the Total Polyphenolic Compound Content (TPC)

The total polyphenolic content (TPC) was determined by following a modified Folin-Ciocalteu (FC) method [14]. The quantity of polyphenols is expressed as mg gallic acid equivalents (GAE) per g of dry weight (d.w.).

### 3.6. Ability to Scavenge DPPH

Measurement of antiradical activity was carried out using DPPH radical scavenging, according to the modified method of Burda and Oleszek [14]. Absorbance was measured at 517 nm wavelength, every 5 min for 30 min by means of a spectrophotometer (Genesys UV-VIS, Thermo Scientific, Waltham, MA, USA). This method allows the monitoring of changes of absorbance over time and defines when the plateau is reached. The free radical scavenging potential of the extracts was figured by employing the following formula:$$\%\mathrm{RSA}=\left[\frac{(A_{0}-A_{1)}}{A_{0}}\right]\times 100$$

A_0_—the absorbance of the sample except tested extracts,

A_1_—the absorbance of the sample with tested extracts.

### 3.7. In Vitro Two-Stage Digestion Model

We applied a static in vitro two-stage (gastric and duodenal) digestion model according to Seraglio et al. [12] with slightly modifications. In order to develop the gastric digestion phase, 1.632 g of each sample was homogenized, mixed with 5.84 mL of gastric solution and stirred for 4 min. To the mixture, 2.32 mL of hydrochloric acid at pH 2.5 ± 0.2 was added. Next, the samples were incubated in a water bath with shaking for two hours (37 °C, 100 rpm). The samples were then centrifuged (10 min, 8000 rpm) and the supernatant were collected for further analysis. The samples were refrigerated at −20 °C for 24 h until the planned analyses were performed.

Duodenal digestion was assessed in the same way as gastric digestion. After incubation, 0.09 mL of 1 mol/L sodium bicarbonate (to increase the pH to 5.5) and 2.26 mL of duodenal solution were added to each sample; the samples were then energetically stirred for 1 min. In the next step, 0.72 mL of sodium bicarbonate solution for adjustment to pH 6.7 ± 0.2 was added to each stock. The samples were then incubated in a water bath with shaking for 2 h (37 °C, 100 rpm). After centrifugalizing (10 min, 8000 rpm), the supernatant was analyzed. The storage conditions were the same as for the gastric step.

The simulated gastric juice was fixed as follows: 0.16 g of pepsin was dissolved in 0.35 mL of 12 M hydrochloric acid, which was made up to 50 mL with ultra-pure water. The duodenal solution was prepared by combining 0.25 g pancreatin with 0.047 g of sodium glycodeoxycholate, 0.0505 g of sodium taurocholate, and 0.029 g of sodium taurodeoxycholate dissolved in 0.25 mL of 0.5 M sodium bicarbonate in 25 mL ultra-pure water.

### 3.8. Moisture

Moisture was determined according to the modified method of Pontieri et al. [4]. A ceramic capsule was accurately weighed after complete desiccation at 100 °C in vacuum-packed conditions using an oven (ISCO mod. NSV9035, Milan, Italy) and left at room temperature in a silica gel dryer. Next, the sorghum sample was inserted into the ceramic capsule. The humidity was removed from the sample, by keeping it in the same temperature and pressure conditions for about five hours, until a constant weight was achieved. The moisture content was measured on the basis of weight loss.

### 3.9. Ash

In order to measure the total ash, the sorghum samples were incinerated at about 550 °C. It was then placed in a desiccator until it cooled down and weighed when room temperature was reached [51].

### 3.10. Protein Content

Nitrogen content was measured using the Kjeldahl method [52]. Sorghum samples were analyzed with a Mineral Six Digester and an Auto Disteam semi-automatic distilling unit (International PBI, Milan, Italy).

### 3.11. Lipid Content

Total lipid content was assessed according to Pontieri et al. [4] with modifications. Sorghum samples were ground with liquid nitrogen using a mortar and pestle, lyophilized with the FTS-System Flex-DryTM equipment and then extracted using a Soxhlet apparatus with chloroform for 4 h. Extracts were completely dried to obtain the crude extracts, which were subsequently weighed to determine the amount of extracted fat.

### 3.12. Carbohydrates

Carbohydrate content was ascertained by subtraction, as the amount of material left after accounting for moisture, ash, protein, and fat content [53].

### 3.13. Fiber

Fiber was determined according to the AOAC [54] method.

### 3.14. Statistical Analysis

All the measurements were replicated at least three times; results were the means of these replicated values and standard deviations (SD). Statistical analysis with ANOVA (Statistica 13.0, StatSoft Inc., Tulsa, OK, USA) was applied to determine the significance of differences at α = 0.05; multi-factor analysis of variance and the Tukey test were also carried out. Pearson’s correlation coefficient values and their significance were evaluated at 0.05 for the tested characteristics. Principal component analysis (PCA), analysis of variance, and the determination of correlations were performed at a significance level of α = 0.05. The principal component analysis was employed to determine the relationships between the total content of polyphenolic compounds, flavonoids, and free phenolic acids in pasta, before and after two-stage digestion and with or without the sorghum addition. The PCA data matrix for the statistical analysis of the results had 6 columns (names of the compounds) and 6 rows (type of pasta before and after two-stage digestion and with the sorghum addition). The input matrix was scaled automatically. The optimal number of principal components obtained in the analysis was determined based on the Cattel criterion.

## 4. Conclusions

The current research provides valuable information on nutrient composition of sorghum and on the unique health benefits of this grain consumption.

The results of the experiments revealed that sorghum is a rich source of phenolics (3.620 mg GAE/g d.w) and flavonoids (0.536 mg GAE/g d.w). The findings also demonstrate the relatively high radical scavenger activity of the extracts against DPPH (63.240%). The following phenolic acids were identified in sorghum grain: protocatechuic, *p*-OH-benzoic, vanillic, syringic, caffeic, and p-coumaric. We observed that the total content of polyphenols decreased after gastric digestion of sorghum and slightly increased after duodenal digestion (compared with the gastric digestion phase). Moreover, the content of flavonoids decreased significantly after the first stage of digestion, while the antioxidant properties increased after the first stage of digestion and slightly decreased after the second stage.

The research results showed that even a small addition of sorghum caused an in-crease in the polyphenol content (including flavonoids and free phenolic acids) in pasta, as well as an increase in antioxidant activity.

The digestion of polyphenolic compounds in sorghum is completely different from that in pasta—both in those with and without the addition of sorghum. For pasta, total polyphenol and flavonoid content, and (especially) free radical scavenging properties decrease after each stage of digestion.

Our work is the first to comprehensively investigate the in vitro digestion of sorghum and pasta enriched with sorghum. Plant products have diverse composition and are often eaten in conjunction with other foods. Our results clearly show that the food matrix ingredients can modulate the bioaccessibility and stability of phytochemicals.

Owing to the fact that the possible effectiveness of plant metabolites for human health is primarily determined by the bioavailability of these molecules, continuous updating of the state of knowledge about the breakdown of food ingredients during digestion is significant.

## Figures and Tables

**Figure 1 molecules-28-01706-f001:**
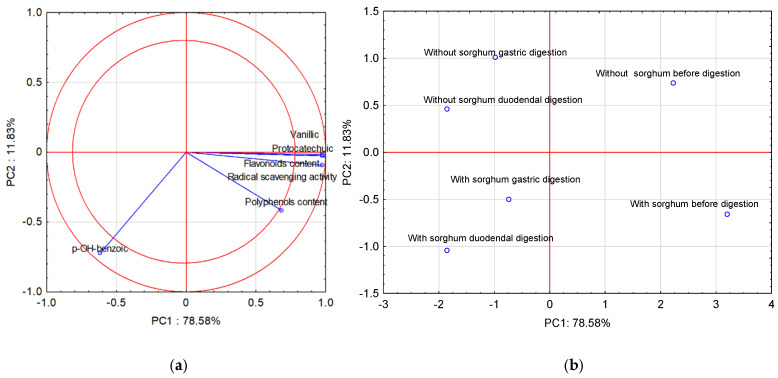
Projection of variables: total content of polyphenolic compounds and flavonoids content of free phenolic acids in pasta on the PC1 and PC2 scores plot—(**a**); projection of cases characterizing the pasta before and after two-stage digestion and with the sorghum addition on the PC1 and PC2 loadings plot—(**b**).

**Figure 2 molecules-28-01706-f002:**
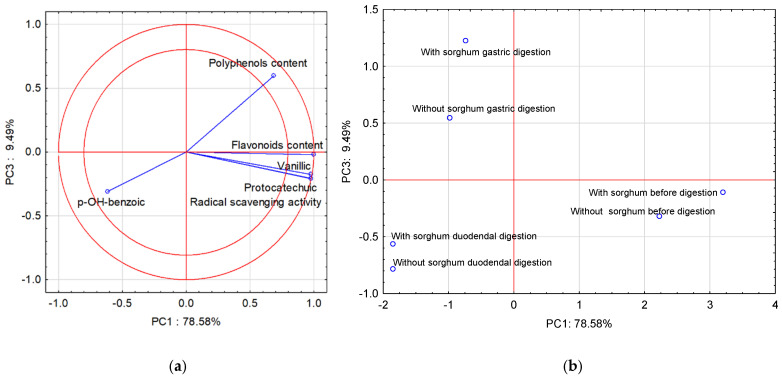
Projection of variables: total content of polyphenolic compounds and flavonoids content of free phenolic acids in pasta on the PC1 and PC3 scores plot—(**a**); projection of cases characterizing the pasta before and after two-stage digestion and with the sorghum addition on the PC1 and PC3 loadings plot—(**b**).

**Figure 3 molecules-28-01706-f003:**
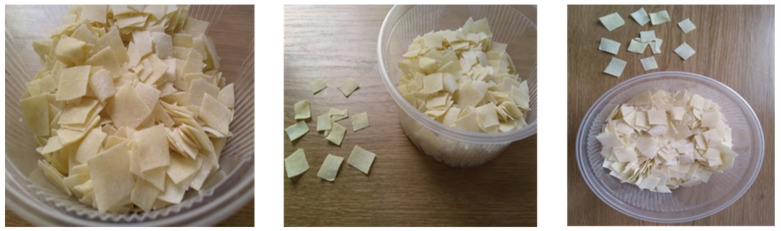
Pasta samples.

**Figure 4 molecules-28-01706-f004:**
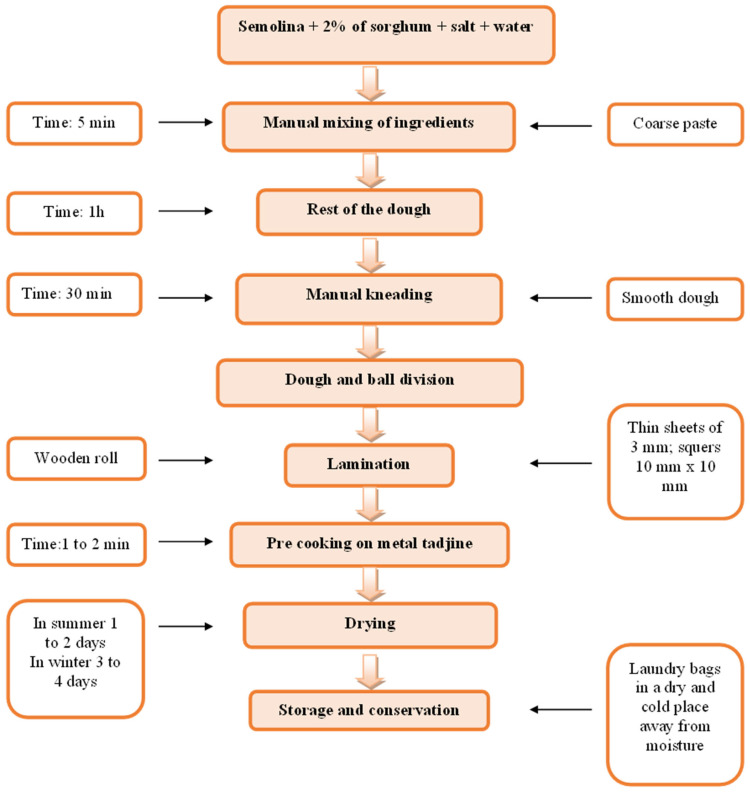
The main stages of the manufacture of pasta.

**Table 1 molecules-28-01706-t001:** Nutritional values of sorghum grain (red variety).

Parameter	Results (%)	Recommended Daily Intake (RDI) (g/Day) [4]
Moisture	8.96 ± 0.07	-
Ash	2.96 ± 0.03	-
Total proteins	8.15 ± 0.12	50
Fats	3.17 ± 0.03	70
Total carbohydrates	48.49 ± 0.76	260
Fiber	28.27 ± 0.23	14 g per 1000 calories of food

**Table 2 molecules-28-01706-t002:** Total content of polyphenolic compounds and flavonoids (TPC mg GAE/g d.w. of extract), as well as radical scavenging activity [%] of sorghum before and after two-stage digestion (*n* = 3; mean ± SD).

Parameters	Before Digestion	Gastric Digestion	Duodenal Digestion
Polyphenol content	3.620 ^a^ ± 0.012	2.991 ^b^ ± 0.056	3.176 ^c^ ± 0.112
Flavonoid content	0.536 ^a^ ± 0.003	0.055 ^b^ ± 0.001	0.061 ^c^ ± 0.000
Radical scavenging activity	63.240 ^a^ ± 1.020	70.117 ^b^ ± 1.321	67.925 ^c^ ± 0.023

Identical letters indicate no significant differences between the obtained results. Different letters (a, b, c) indicate significant differences between the results obtained for the compared compounds and cases.

**Table 3 molecules-28-01706-t003:** The free phenolic acid content (µg/g d.w.) in sorghum before and after two-stage digestion (*n* = 3; mean ± SD).

Phenolic Acid	Before Digestion	Gastric Digestion	Duodenal Digestion
Protocatechuic	3.663 ^a^ ± 0.032	4.314 ^b^ ± 0.024	3.717 ^a^ ± 0.056
*p*-OH-benzoic	6.947 ^a^ ± 0.124	7.790 ^b^ ± 0.211	7.851 ^b^ ± 0.003
Vanillic	1.612 ^a^ ± 0.009	2.661 ^b^ ± 0.014	2.675 ^b^ ± 0.021
Caffeic	0.568 ^a^ ± 0.005	0.813 ^b^ ± 0.000	1.244 ^c^ ± 0.047
Syringic	0.107 ^a^ ± 0.003	0.355 ^b^ ± 0.001	0.376 ^c^ ± 0.005
*p*-coumaric	1.461 ^a^ ± 0.045	3.966 ^b^ ± 0.075	3.973 ^b^ ± 0.103
Sum	14.358	19.899	19.837

Identical letters indicate no significant differences between the obtained results. Different letters (a, b, c) indicate significant differences between the results obtained for the compared compounds and cases.

**Table 4 molecules-28-01706-t004:** Total content of polyphenolic compounds and flavonoids (mg GAE/g d.w.), as well as radical scavenging activity [%] of pasta before and after two-stage digestion (*n* = 3; mean ± SD).

	Parameters	Before Digestion	Gastric Digestion	Duodenal Digestion
Pasta without sorghum	Polyphenol content	0.659 ^a^ ± 0.007	0.623 ^b^ ± 0.005	0.542 ^c^ ± 0.003
	Flavonoid content	0.064 ^a^ ± 0.001	0.011 ^b^ ± 0.000	-
	Radical scavenging	30.251 ^a^ ± 0.265	5.126 ^b^ ± 0.109	4.041 ^c^ ± 0.075
Pasta enriched with sorghum	Polyphenol content	0.748 ^d^ ± 0.032	0.728 ^e^ ± 0.003	0.613 ^f^ ± 0.004
	Flavonoid content	0.080 ^d^ ± 0.000	0.017 ^e^ ± 0.001	0.001 ^f^ ± 0.000
	Radical scavenging	41.999 ^d^ ± 0.897	5.323 ^e^ ± 0.070	5.284 ^f^ ± 0.236

Identical letters indicate no significant differences between the obtained results. Different letters (a, b, c, d, e, f) indicate significant differences between the results obtained for the compared compounds and cases.

**Table 5 molecules-28-01706-t005:** The content of free phenolic acids (µg/g d.w.) in pasta before and after two-stage digestion (*n* = 3; mean ± SD).

	Phenolic Acid	Before Digestion	Gastric Digestion	Duodenal Digestion
Pasta without sorghum	Protocatechuic	0.051 ^a^ ± 0.000	-	-
	*p*-OH-benzoic	0.112 ^a^ ± 0.002	0.121 ^b^ ± 0.004	0.158 ^c^ ± 0.003
	Vanillic	0.132 ^a^ ± 0.004	-	-
	Sum	0.295	0.121	0.158
Pasta enriched with sorghum	Protocatechuic	0.060 ^d^ ± 0.001	-	-
	*p*-OH-benzoic	0.136 ^d^ ± 0.009	0.149 ^e^ ± 0.006	0.193 ^f^ ± 0.001
	Vanillic	0.160 ^d^ ± 0.003	-	-
	Sum	0.359	0.149	0.193

Identical letters indicate no significant differences between the obtained results. Different letters (a, b, c, d, e, f) indicate significant differences between the results obtained for the compared compounds and cases.

**Table 6 molecules-28-01706-t006:** Pearson’s correlation coefficient at α < 0.05.

Sample	Compound	Total Polyphenols	Free Phenolic Acid	Radical Scavenging Capacity
Sorghum	Total flavonoids	0.962	−0.999	−0.953
	Total polyphenols		−0.966	−0.999
	Free phenolic acids			0.953
Pasta without sorghum	Total flavonoids	0.836	0.934	0.992
	Total polyphenols		0.585	0.761
	Free phenolic acids			0.971
Pasta enriched with sorghum	Total flavonoids	0.754	0.923	0.982
	Total polyphenols		0.442	0.615
	Free phenolic acids			0.979

## Data Availability

Not applicable.

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
