# Peer review of "The Effect of In Vitro Digestion on Polyphenolic Compounds and Antioxidant Properties of Sorghum (Sorghum bicolor (L.) Moench) and Sorghum-Enriched Pasta"

_molecules, 2023, doi:10.3390/molecules28041706_

Round 1
Reviewer 1 Report
I have included the comments on the attached manuscript:
1. The manuscript needs an English editor
2. I made changes to the title
3. The study is very important in the field and presents some new and interesting information especially on bioavailability of phenolic compounds.
4. However, PCA should be employed to further understand this data. Comparisons between p0henolic compounds in plain sorghum and that infused with pasta needs to be discussed.
5. The abstract needs to be overhauled to capture the results and conclusions.

Author Response
I have included the comments on the attached manuscript:
The authors would like to thank the Reviewer for valuable comments which have helped to improve the quality of the manuscript. We hope that the revisions in the manuscript and accompanying responses will be sufficient to make our manuscript suitable for publication. We have made all the changes suggested in the Reviewer's comments in the text.
- The manuscript needs an English editor
Thank you for valuable comment. The manuscript has been improved towards grammar and stylistics by native speaker Jack Stanley Dunster from Canada (Language Editor of Current Issues in Pharmacy and Medical Sciences), who has many years of experience in this type of work.
- I made changes to the title
We would like to thank the Reviewer for the changes in the title.
- The study is very important in the field and presents some new and interesting information especially on bioavailability of phenolic compounds. However, PCA should be employed to further understand this data. Comparisons between p0henolic compounds in plain sorghum and that infused with pasta needs to be discussed.
Thank you for your suggestion. Principal component analysis (PCA) was performed for the obtained research results. The following description of the obtained correlations and relationships with figures has been added to the manuscript:
Performing the principal components analysis (PCA) allowed to obtain five new variables, and the first three principal components explain 99.90% of the variability of the system. Figure 1a shows the projection of the variables on planes PC1 (78.58%) and PC2 (11.83%), which describe the dependencies at 90.41%.
A strong positive correlation was found between: vanillic acid, protocatechuic acid, Flavonoids content and radical scavenging activity. The correlation between these parameters and p-OH-benzoic acid was strong and negative. The correlation between: vanillic acid, protocatechuic acid, flavonoids content, radical scavenging activity and polyphenols content is positive but weak. In turn, there is no correlation between polyphenols content and p-OH-benzoic. All compounds found in the two-circle area strongly affect the determination ability of the pasta before and after two-stage digestion and with the sorghum addition. Figure 1b shows cases of pasta before and after two-stage digestion and with the sorghum addition. Positive PC1 values describe cases before digestion, and negative PC1 values describe cases after two-stage digestion. In turn, the second principal component (PC2) in Figure 1b describes the case of pasta with the sorghum addition. Positive PC2 values describe pasta without sorghum addition, and negative PC2 values describe cases of pasta with sorghum addition.
Due to the fact that PC3 is close to the PC2 value, the third principal component (PC3) was also used in the interpretation of the analysis results (PCA). Figure 2a shows the projection of the variables on planes PC1 (78.58%) and PC3 (9.49%), which describe the dependencies at 88.07%.
A strong positive correlation was found between: vanillic acid, protocatechuic acid, flavonoids content and radical scavenging activity. The correlation between: vanillic, protocatechuic, flavonoids content, radical scavenging activity and polyphenols content is positive but weak. The correlation between these parameters and p-OH-benzoic acid was weak and negative. In turn, the correlation between polyphenols content and p-OH-benzoic acid is strong and negative. All compounds found in the two-circle area strongly affect the determination ability of the pasta before and after two-stage digestion and with the sorghum addition. Figure 2b shows cases of pasta before and after two-stage digestion and with the sorghum addition. Positive PC1 values describe cases before digestion, and negative PC1 values describe cases after two-stage digestion. In turn, the third principal component (PC3) in Figure 2b describes the case of after two-stage digestion. Positive PC3 values describe gastric digestion and negative PC3 values describe cases of duodendal digestion.
Basic information about the criteria used in the PCA analysis has also been added in section 3. Materials and Methods in subchapter 3.14. Statistical analysis:
Statistical analysis with ANOVA (Statistica 13.0, StatSoft Inc., Tulsa, OK, USA) was applied to determine the significance of differences at α = 0.05, multi-factor analysis of variance and the Tukey test were carried out. Pearson’s correlation coefficient and their significance were evaluated at 0.05 for the tested characteristics. Principal component analysis (PCA), analysis of variance, and the determination of correlations were performed at a significance level of α = 0.05. The principal component analysis was employed to determine the relationships between the total content of polyphenolic compounds, flavonoids content of free phenolic acids in pasta before and after two-stage digestion and with the sorghum addition. The PCA data matrix for the statistical analysis of the results had 6 columns (names of the compounds) and 6 rows (type of pasta before and after two-stage digestion and with the sorghum addition). The input matrix was scaled automatically. The optimal number of principal components obtained in the analysis was determined based on the Cattel criterion.
- The abstract needs to be overhauled to capture the results and conclusions.
Thank you for valuable comment. The abstract has been completely changed.
Other Reviewer’s comments:
Section 2.3: I would be more comfortable if the authors discussed in comparison with members of the Poaceae family or where sorghum belongs.
Unfortunately, there are very few studies on the in vitro digestion of cereals. Therefore, we supplemented the section with the results of in vitro digestion studies of corn (Poaceae family) snacks enriched with wild garlic, which we published in a previous paper.
Section 2.3: How do these values compare with those of rice, wheat and maize... these crops are staples throughput the world?
We have added the information suggested by the Reviewer to section 2.3
Reviewer 2 Report
Title Effect of the In Vitro Digestion on the Content of Polyphenolic Compounds and Antioxidant Properties of Sorghum (Sorghum 3 bicolor (L.) Moench) and Pasta Enriched With Sorghum Comments
Comments: The manuscript needs minor revisions before publication
Ø Your aim was to investigate the content of polyphenolic compounds, phenolic acids, flavonoids and the anti-radical properties of sorghum and traditional Algerian-enriched sorghum pasta. What is new from the previous work? What is the advantage to examine the content before and after digestion?
Ø Line 32. The keywords’ writing style is not correct. and too-long keywords. Write the first word of each keyword with capital letters.
Ø The abstract should contain your main result. Please add the value of each result for example total flavonoid content, Antioxidant activity, and others
Ø The abstract should contain the conclusion and future perspective.
Ø The introduction section does not clearly discuss the statement of the problem and the gaps. please correct it.
Ø Line 98. I did not observe your explanation. why the higher fiber content found in your sorghum compared to the work of Pontieri et al. please include the discussion in your result and discussion part
Ø Line 135. Change the word “We utilized..” It is an individualization word not recommended
Ø Line 95, 171, 458 significant is the statical word “ significantly (p < 0.05)” but all of your data does not show the statically regarding significant information
Ø Your conclusion is a little big please reduce and must contain the main finding with a conclusion and recommendation. No need to write the aim of the study in the conclusion line 446-450
Author Response
Title Effect of the In Vitro Digestion on the Content of Polyphenolic Compounds and Antioxidant Properties of Sorghum (Sorghum 3 bicolor (L.) Moench) and Pasta Enriched With Sorghum Comments. Comments: The manuscript needs minor revisions before publication
The authors would like to thank the Reviewer for valuable comments which have helped to improve the quality of the manuscript. We hope that the revisions in the manuscript and accompanying responses will be sufficient to make our manuscript suitable for publication.
Ø Your aim was to investigate the content of polyphenolic compounds, phenolic acids, flavonoids and the anti-radical properties of sorghum and traditional Algerian-enriched sorghum pasta. What is new from the previous work? What is the advantage to examine the content before and after digestion?
To date, no one has tested the digestion of sorghum or the digestion of products enriched with sorghum.The gastrointestinal digestion of foods significantly influences the bioaccessibility of biologically active compounds such as polyphenols. Since plant foods are diverse in composition or eaten in conjunction with other foods, food bolus constituents can modulate the bioaccessibility and stability of phytochemicals. It should be remembered that before polyphenols can be used in the body, they are modified within the digestive tract in order to be assimilated. For this reason, studies that are based only on the de-termination of the concentration of phytonutrients in food products do not provide complete information on the actual bioavailability of these components. In order to obtain more accurate data on the level and activity of tested ingredients after ingestion and digestion in the in vivo digestive tract, it seems more justified to use in vitro diges-tion experiments. Thus, the aim of this study was to determine the content of polyphenolic compounds, flavonoids, individual phenolic acids, and antiradical prop-erties of sorghum and pasta enriched with sorghum before and after in vitro simulated gastrointestinal digestion.
Ø Line 32. The keywords’ writing style is not correct. and too-long keywords. Write the first word of each keyword with capital letters.
Thank you for comment. Keywords style has been corrected.
Ø The abstract should contain your main result. Please add the value of each result for example total flavonoid content, Antioxidant activity, and others. The abstract should contain the conclusion and future perspective.
Thank you for valuable comment. The abstract has been completely changed.
Ø The introduction section does not clearly discuss the statement of the problem and the gaps. please correct it.
Thank you for valuable comment. We supplemented the introduction with the statement of the problem and the gaps.
Ø Line 98. I did not observe your explanation. why the higher fiber content found in your sorghum compared to the work of Pontieri et al. please include the discussion in your result and discussion part
Thank you for your suggestion. We added to paragraph 2.1. discussion about the fiber content.
Ø Line 135. Change the word “We utilized..” It is an individualization word not recommended
Thank you for your suggestion. The word ’”we utilized“ has been changed to ”we used”.
Ø Line 95, 171, 458 significant is the statical word “ significantly (p < 0.05)” but all of your data does not show the statically regarding significant information
Thank You for your comment. Statistical analyzes and principal component analysis (PCA) were performed. Basic information about the criteria used in the PCA analysis has also been added in section 3. Materials and Methods in subchapter 3.14. Statistical analysis:
Statistical analysis with ANOVA (Statistica 13.0, StatSoft Inc., Tulsa, OK, USA) was applied to determine the significance of differences at α = 0.05, multi-factor analysis of variance and the Tukey test were carried out. Pearson’s correlation coefficient and their significance were evaluated at 0.05 for the tested characteristics. Principal component analysis (PCA), analysis of variance, and the determination of correlations were performed at a significance level of α = 0.05. The principal component analysis was employed to determine the relationships between the total content of polyphenolic compounds, flavonoids content of free phenolic acids in pasta before and after two-stage digestion and with the sorghum addition. The PCA data matrix for the statistical analysis of the results had 6 columns (names of the compounds) and 6 rows (type of pasta before and after two-stage digestion and with the sorghum addition). The input matrix was scaled automatically. The optimal number of principal components obtained in the analysis was determined based on the Cattel criterion.
The results of the analysis were added to the results in the tables in the form of letters. The same letters indicate no significant differences between the obtained results. Different letters (a, b, c, d, e, f) or (a, b, c) indicate significant differences between the results obtained for the compared compounds and cases.
Ø Your conclusion is a little big please reduce and must contain the main finding with a conclusion and recommendation. No need to write the aim of the study in the conclusion line 446-450
Thank you for valuable comment. The abstract has been completely changed according to Reviwer’s suggestion.